# Potential Use of Nitrogen-Doped Carbon Nanotube Sponges as Payload Carriers Against Malignant Glioma

**DOI:** 10.3390/nano11051244

**Published:** 2021-05-08

**Authors:** Alelí Salazar, Verónica Pérez-de la Cruz, Emilio Muñoz-Sandoval, Víctor Chavarria, María de Lourdes García Morales, Alejandra Espinosa-Bonilla, Julio Sotelo, Anabel Jiménez-Anguiano, Benjamín Pineda

**Affiliations:** 1Graduate Program in Experimental Biology, DCBS, Universidad Autónoma Metropolitana-Iztapalapa, Ciudad de México 09340, Mexico; ajsalazar27@gmail.com; 2Neurosciences Area, Biology of the Reproduction Department, Universidad Autónoma Metropolitana-Iztapalapa, Ciudad de México 09340, Mexico; aja@xanum.uam.mx; 3Neuroimmunology and Neuro-Oncology Unit, Instituto Nacional de Neurología y Neurocirugía (INNN), Mexico City 14269, Mexico; david_vdcy@hotmail.com (V.C.); jsotelo@unam.mx (J.S.); 4Neurobiochemistry and Behaviour Laboratory, Instituto Nacional de Neurología y Neurocirugía (INNN), Mexico City 14269, Mexico; veped@yahoo.com.mx; 5Division of Advanced Materials, IPICYT, San Luis Potosí 78216, Mexico; ems@ipicyt.edu.mx; 6Escuela Nacional de Ciencias Biológicas, Instituto Politécnico Nacional, Mexico City 11340, Mexico; lulugm93255@gmail.com; 7Central de Instrumentación, Posgrado en Inmunología, Escuela Nacional de Ciencias Biológicas, Instituto Politécnico Nacional, Mexico City 11340, Mexico; alecittab@gmail.com

**Keywords:** nitrogen-doped carbon nanotube sponges, carmustine, nanomaterials for delivery, glioblastoma, cytotoxicity

## Abstract

Glioblastoma is the most aggressive brain tumor with a low median survival of 14 months. The only Food and Drug Administration (FDA)-approved treatment for topical delivery of the cancer drug carmustine is Gliadel. However, its use has been associated with several side-effects, mainly provoked by a mass effect. Nitrogen-doped carbon nanotube sponges (N-CNSs) are a new type of nanomaterial exhibiting high biocompatibility, and they are able to load large amounts of hydrophobic drugs, reducing the amount of carriers. This study evaluated the use of N-CNSs as potential carmustine carriers using malignant glioma cell lines. N-CNSs were characterized by nanoparticle tracking analysis and transmission electron microscopy. The biocompatibility of N-CNSs was determined in glioma cell lines and in primary astrocytes. Afterward, N-CNSs were loaded with carmustine (1:10 *w*/*w*), and the drug and liberation efficiency, as well as cytotoxicity induction, were determined. N-CNSs presented a homogeneous size distribution formed by round nanotubes, without induced cytotoxicity, at concentrations below 40 µg/mL. The N-CNSs loaded with carmustine exhibited a continuous kinetic release of carmustine with a maximum release after 72 h. The cytotoxic effect of N-CNSs loaded with carmustine was similar to that of carmustine alone. The results demonstrated that N-CNSs are a biocompatible nanostructure that could be used as carriers for the tumoral load of large amounts of chemotherapeutic agents.

## 1. Introduction

Glioblastoma (GBM) is the most frequent and aggressive brain tumor. It has remained incurable despite current advances in diagnosis and treatment. GBM presents an incidence rate of 5–8 cases per 100,000 per year [1]. There are currently two main clinical practices approved by the FDA to treat these types of neoplasms. The first is based on the Stupp protocol, which consists of surgery followed by chemotherapy with temozolomide (TMZ) concomitantly with radiotherapy, which increases the median survival time to 14.6 months; the second is based on surgery and the local administration of Gliadel, a polymer wafer containing bis-chloro-ethyl-nitrosourea, also known as carmustine (BiCNU), followed by radiotherapy with a median survival of 13.9 months [2]. Recently, various studies have shown evidence that both protocols can be combined [3]. 

Although TMZ increases the survival of GBM patients, its low tissue penetration through the blood–brain barrier and its systemic administration potentially reduce the effective concentration in the tumor mass, limiting its safety profile [4]. Gliadel was designed to overcome some of these concerns in delivering the chemotherapeutic BiCNU in the surgical bed, achieving a local intratumoral concentration 100-fold greater than that achieved by the systemic route [5], thus limiting systemic toxicity. Gliadel wafers consist of polymer discs of 1.45 cm in diameter, 1 mm in thickness, and 200 mg in weight, and the total number of wafers per treatment consists of around 6–8 wafers (with an approximate cost of USD 26,325) [6]. Although Gliadel has been shown to increase overall survival by 2.3 months, this treatment has also shown a high incidence of side-effects related to the compression caused by the foreign object over the surrounding brain structures within the skull, inducing injury, a phenomenon known as “mass effect” (seizures, cerebral edema, and infection), usually two weeks post-implantation by surgery [7]. Due to these limitations, it is necessary to find new forms of drug delivery for the treatment of GBM, exposing the tumor to high concentrations of the drug, while minimizing systemic exposure, which would finally lead to reduced adverse drug-related effects and increased overall survival.

Nitrogen-doped carbon nanotube sponges (N-CNSs) are carbon sponges formed by carbon nanotubes doped with nitrogen that possess superhydrophobic, oil-absorbent, and electroactive properties, as well as good absorption capacities with a load capacity of up to 100 times its own weight [8]; all of these characteristics prove useful as appropriate carriers of various chemotherapeutic agents for local delivery, mainly for solid tumors in which the side-effects induced by chemotherapeutics significantly reduce the quality of life of patients. To overcome the side-effects induced by the “mass effect” in the local drug delivery to tumors, biocompatible and large-payload-carrying carriers of hydrophobic substances have been designed. In this study, we evaluated the N-CNSs, previously synthesized by our group [8], as nanocarriers loaded with BiCNU. The morphology, size, release profile, and toxicity induced by the carrier and by the carrier loaded with BiCNU were characterized in various malignant glioma cell lines and in a primary cell culture of astrocytes as a healthy counterpart.

## 2. Materials and Methods

### 2.1. Materials

Dulbecco’s modified Eagle’s culture medium (DMEM) was obtained from GIBCO Invitrogen. Penicillin, streptomycin, 3-(4,5-dimethyltiazol-2-yl)-2,5-diphenyltetrazolium bromide (MTT), and carmustine (bis-chloroethylnitrosourea, BiCNU) were obtained from Sigma Aldrich. BiCNU was dissolved in 100% ethanol (Sigma Chemical Co., St. Louis, MO, USA) for the assays. N-CNSs were manufactured by the pyrolysis of a solution of ferrocene (Fe(C_5_H_5_)_2_) and benzylamine (C_6_H_5_CH_2_NH_2_) via aerosol-assisted chemical vapor deposition (AACVD) using two different sprayers (RBI instrumentation), reported elsewhere [9]. The individual carbon nanotubes building the carbon sponge were multiwalled (MWCNTs).

### 2.2. Characterization of Nitrogen-Doped Carbon Nanotube Sponges (N-CNSs) by Nanoparticle Tracking Analysis (NTA) 

The size distribution of N-CNSs was determined via dynamic light scattering with a Malvern Zetasizer NS300 instrument (NanoSight, Amesbury, UK). All samples were diluted in PBS to a final volume of 1 mL. Ideal measurement concentrations were found by pre-testing the ideal particle per frame value (20–100 particles/frame). A sample of the N-CNSs was placed on the instrument equipped with a blue laser (488 nm) and a complementary metal–oxide–semiconductor (CMOS) camera, which allowed the fast and automatic analysis of the size distribution and concentration of nanoparticles, from 10 to 2000 nm in diameter, according to the configuration of the instrument and the type of sample. Preparations were measured in quintuplicate (temperature, 21.2–21.4 °C; viscosity, 0.968–0.971 centipoise (cP)) (water) for 10 s. The software used for the capture and analysis of data was NTA 3.2 Dev Build 3.2.16.

### 2.3. Characterization of Nitrogen-Doped Carbon Nanotube Sponges by Transmission Electron Microscopy (TEM)

The morphology of the N-CNSs was observed under TEM (FEI Tecnai G 220 S-Twin, Hillsboro, OR, USA). One drop of suspension was deposited on a carbon-coated copper grid loaded with 5 µL of the N-CNSs suspended in Milli-Q water, and it was allowed to dry for contrast enhancement with 1.5% uranyl acetate. Images were acquired using a Gatan OneView 4K camera mounted on a Jem-2100Plus (Jeol, Tokyo, Japan) operating at 200 kV. Samples were analyzed with ImageJ software (NIH).

### 2.4. Cellular Uptake Evaluation of Nitrogen-Doped Carbon Nanotube Sponges 

To estimate the cellular uptake of N-CNSs in rat astrocytes and C6, RG2, and U87 cells, cells were cultured in 24-well plates (1 × 10^5^ cells) and were treated with different concentrations of N-CNSs (10, 20, 30, 40, 50, 60, 70, 80, 90, and 100 µg/mL) for 48 h. The cells were harvested by trypsin-EDTA, were washed twice with PBS, and were resuspended in PBS for flow cytometry analysis (10,000 cumulative events, analyzed by CELLQUEST software). The mean increases in size (forward scatter, FSC) and internal complexity (side scatter, SSC) were determined using FlowJo v 10. Software. Results were plotted as the mean ± SD of three different experiments in triplicate. 

### 2.5. Loading Carmustine into Nitrogen-Doped Carbon Nanotube Sponges 

The solvent evaporation technique is one of the most used methods to prepare drug-loaded polymeric systems for pharmaceutical formulations. To load BiCNU, 10 µg of N-CNSs were mixed with BiCNU (230 µM previously prepared in 100% ethanol) by stiring for 24 h. Then, the drug-loaded N-CNSs (drug–carrier ratio of 10:1) was dried at room temperature into a sterile flow hood. The amount of the loaded drug was analyzed by using a UV spectrophotometer (Nanodrop, ThermoFisher Scientific, Wilmington, DE USA) at 260 nm to ascertain the BiCNU content in nanosponges. 

### 2.6. Carmustine Liberation Efficiency from Nitrogen-Doped Carbon Nanotube Sponges

First, 10 µg of loaded N-CNSs was placed in 100 µL of DMEM medium or artificial cerebrospinal fluid (CSF) and at specific time intervals (0, 0.25, 0.5, 1, 3, 5, 8, 12, 24, 48, and 72 h); a sampling of dissolution was carried out with amounts of 5 µL. After each sampling, DMEM medium or CSF was added into the tube to keep the total volume of solution constant. The absorbance of samples was read with a UV-Vis nanodrop spectrophotometer at a wavelength of 260 nm. A standard curve was plotted for this purpose and was used to determine drug concentrations. The linearity of BiCNU concentrations in the medium solution (DMEM and artificial CSF) with respect to UV-absorbance was evaluated by the construction of calibration curves corresponding to peak areas of 8 drug concentrations in the range from 0 to 260 μM with 3 replicates by linear regression. Finally, the release rate of drug versus time was determined.

### 2.7. Cytotoxic Effects of N-CNSs and N-CNSs Loaded with Carmustine in Cell Cultures

Primary astrocytes in cell culture were used as nonmalignant control cells as the origin of GBM tumors from transformed astrocytes. Astrocytes were isolated from 3-day-old Fisher rats, as previously described [10]. The care and use of all experimental animals were performed in accordance with institutional ethical guidelines. RG2, C6 (rat malignant glioma), and U87 (human GBM) tumor cell lines were acquired from the American Type Culture Collection (ATCC, Manassas, VA, USA). The cells were cultured with DMEM (GIBCO BRL, Grand Island, NY, USA) and were supplemented with 10% bovine fetal serum (GIBCO BRL, Grand Island, NY, USA), 4 mM of glutamine, and 100 U/mL of penicillin–streptomycin, cultured under sterile conditions at 37 °C in a humid atmosphere with 5% CO_2_. The N-CNS cytotoxicity was determined in rat astrocytes and C6, RG2, and U87 cells. Briefly, cells were cultured in 96-well plates (1 × 10^4^ cells) and were treated with different concentrations of N-CNSs (10, 20, 30, 40, 50, 60, 70, 80, 90, and 100 µg/mL). After determining the N-CNS cytotoxicity, 1 × 10^4^ RG2 glioma cells were cultured in 96-well plates, and after 24 h, cells were treated with eight N-CNS serial dilutions (1:2, from 10 µg), BiCNU (initial concentration: 230 µM) or N-CNSs loaded with BiCNU (10 µg loaded with 230 µM BiCNU), or with 1% of ethanol as a control group. After 24, 48, and 72 h, the cell viability was determined via MTT assay. Briefly, after treatment, the medium was removed, and the cells were washed with PBS; then, 100 µL of MTT (5 mg/mL in PBS) was added to each well. Cells were incubated for 4 h at 37 °C; then, the medium was removed, and the blue formazan product was eluted with acidic-isopropanol. Next, the acidic-isopropanol containing formazan was centrifuged at 14,000× *g* to remove N-CNSs. The supernatant was collected and quantified. Quantification of formazan was determined by the optical density at a wavelength of 570 nm in a plate reader (EON; BioTek). The results were expressed as % of viability in relation to the control values. The drug concentration or the N-CNSs loaded with BiCNU causing 50% inhibition (IC_50_) was calculated using the GraphPad Prism software V. 8.0.2.

### 2.8. Statistical Analysis

Quantitative data were presented as means ± standard deviation (SD). The statistical significance was analyzed using the Student’s t-test with a p value less than 0.05 (*p* < 0.05) indicating significance. All the tests were carried out using GraphPad Prism v. 8.4.3 software.

## 3. Results

### 3.1. Characterization of N-CNSs

The particle size distributions (PSDs) of N-CNSs were analyzed using the nanoparticle size distribution (NTA). The size distribution profiles of N-CNSs produced major peaks at 254.1 nm, and the mean size of the N-CNS diameter was 241.7 ± 95.1 nm (Figure 1). 

Additionally, the TEM images shown in Figure 2 corroborate that the CNTSs, fabricated by the two-furnace-configuration AACVD method, constituted curved and entangled nanotubes (Figure 2A) exhibiting CNT junctions (Figure 2B) and a porous structure (Figure 2C). Figure 2A shows the individual CNTs building the N-CNSs. Figure 2B shows nanotubes with a “spaghetti” shape and nanoribbons with rolled-up edges, and straight-angles formed by carbon nanotubes also are observed (T-junction). Figure 2C shows a nano sponge that was not separated during sample preparation. The size determined by TEM was consistent with the NTA Nanosight; thus, each analysis showed similar results in size distribution.

### 3.2. N-CNSs Increase Cell Granularity in a Dose-Dependent Manner

The following step consisted of the evaluation of the cell uptake of N-CNSs, which was determined through the change in cellular granularity. N-CNSs were efficiently endocytosed by astrocytic cells, including primary astrocytes and different malignant glioma cells (Figure 3). The number of N-CNSs endocytosed by these cells was proportional to the number of N-CNSs in contact with them. Flow cytometry was applied to estimate the cellular uptake of N-CNSs in C6, U87, RG2 glioma cells, and primary astrocytes through side scatter (SSC) analysis to determine the increase in the internal complexity expressed as granularity (Figure 3A). The increase in granularity was proportional to the number of N-CNSs placed in contact with the cells (Figure 3B). In addition, N-CNSs were observed inside the cells into the cytoplasm throughout the all-glioma cells and primary astrocytes, indicating their potent internalization after 48 h of incubation (Figure 3C). No morphological changes were observed, only a small dot pattern inside the cells.

### 3.3. Effect of N-CNSs on Cellular Viability

Cell viability was assessed via MTT assay in astrocytes, C6, U87, and RG2 glioma cells treated with different concentrations of N-CNSs (0–100 µg/mL). The addition of N-CNSs to the culture medium did not affect the cell viability at 24 and 48 h (data not shown). However, N-CNSs increased the percentage of cell death significantly from 40 µg/mL for astrocytes and U87 cells (23 ± 10 and 24 ± 6 vs. control, respectively), from 60 µg/mL for RG2 (17.7 ± 5 vs. control), and from 100 µg/mL for C6 (11.6 ± 5 vs. control) cells. (Figure 4). The effect observed in all culture cells tested, except for C6 cells, was concentration-dependent. Although cell viability was affected after 72 h of treatment, the CC_50_ of N-CNSs was >100 µg/mL for all the cells used. Based on the results obtained after 72 h of treatment, 10 µg of N-CNSs was used for subsequent experiments.

### 3.4. Carmustine Release Profile of N-CNSs Loaded with Carmustine (N-CNSs-BiCNU)

Figure 5 shows the BiCNU release profiles from N-CNSs in DMEM and CSF. The drug was released from N-CNSs at a sustained rate. The N-CNSs-BiCNU release was slow in both CSF and DMEM; the complete release of the BiCNU in both media occurred after 72 h. A faster release of BiCNU from N-CNSs was seen within the first hour; then, a slow rate of release was observed between 3 and 8 h, continuing with a slow release from 24 to 72 h (completing the full release of BiCNU). These may indicate that the N-CNS structure retains the BiCNU, probably over and inside the N-CNSs.

### 3.5. Cytotoxicity Induction on RG2 Glioma Cells by N-CNSs-BiCNU

The cytotoxicity of the N-CNSs, BiCNU, and BiCNU loaded in N-CNS (N-CNSs-BiCNU) formulations was evaluated and compared in RG2 cells at various concentrations. Figure 6 shows the percentage of cytotoxicity at 24, 48, and 72 h. The cell inhibition rates of BiCNU and N-CNSs-BiCNU conformed to concentration-dependent patterns in all times tested. The IC_50_ values of BiCNU and N-CNSs-BiCNU were similar between them (7.51 ± 1.83 and 3.26 ± 0.86 μM at 24 h; 0.71 ± 0.3 and 0.74 ± 0.4 μM at 48 h, and 0.41 ± 0.19 and 1.46 ± 0.12 μM at 72 h, respectively). There was no induction of cytotoxicity shown by the N-CNSs at any of the times tested. The results demonstrated that the induction of death in RG2 cells was due to the BiCNU toxicity in both cases (BiCNU and N-CNSs-BiCNU treatments) as no relevant differences between the cytotoxicity caused by BiCNU and N-CNSs-BiCNU were documented. Thus, these data also indicate that the carrier used in those experiments did not affect the cytotoxicity capacity of BiCNU in RG2 cells. 

## 4. Discussion

Glioblastoma is a highly invasive and deadly brain tumor, and current therapeutic approaches consist of maximal surgical resection followed by chemotherapy and radiotherapy [11]. Despite the current advances in the diagnosis and treatment of glioblastoma such as molecular subtyping, tailored oncological treatments, including the Stupp regime and carmustine wafers, image-guided surgery, and Glioma-assisted resections, the survival expectancy has only increased discreetly and less than 10% survive more than 3 years with all modalities of treatment [12].

Currently, Gliadel remains the only approved biodegradable agent used for the local delivery of chemotherapeutics, particularly BiCNU, to treat glioblastoma patients, increasing the survival time up to three months as compared to the conventional TMZ regimen [13]. However, treated patients frequently experience side-effects, including edema, an increase in intracranial pressure, seizures, cerebrospinal fluid leakage, intracranial infections, hydrocephalus, cerebral cyst formation, and mass effect [14,15]. Many of these effects could be reduced by the substitution of the payload agent by nonreactive biocompatible nanocarriers with large capacities to be loaded with chemotherapeutic agents, reducing the amount of the carrier while increasing the contact area between the tumor and the chemotherapeutic substance, and consequently reducing the mass effect with similar therapeutic efficiency [16]. Particularly, the N-CNSs could be delivered into the surgical bed, allowing the liberation of the same amount of carmustine and reducing the quantity of the carrier. The release profile observed in our study allowed us to predict that it is possible that N-CNSs could be used as the Gliadel is used. 

The N-CNSs used in this study were formed by coaxial multiwalled carbon nanotubes (MWCNT) produced by the AACVD method, which induced the production of clean and homogenous MWCNTs with a round diameter size of 241.7 ± 95.1 nm and with a rolled porous structure resembling spaghetti figures. This special shape and structure provided it with high hydrophobic oil absorption properties, allowing it to be loaded with large amounts of BiCNU. In contrast to BiCNU wafers [7], a small number of N-CNSs could be used, providing the same bolus release of chemotherapy in situ, reducing the toxicity, and improving the biocompatibility of the nanocarrier. 

Additionally, N-CNSs increase the cell internal complexity in a dose-dependent manner without inducing significant morphological changes. The high biocompatibility of the N-CNSs allows the cellular uptake by glioma and astrocytes without relevant cytotoxicity. The biocompatibility of carbon nanostructures has been previously reported [17]. Our results demonstrate that N-CNSs were noncytotoxic in vitro at concentrations below 40 µg/mL, and the low toxicity induced by N-CNSs was probably due to the high degree of functionalization of the nanomaterial used [8]. The functionalization of these nanomaterials has been shown to significantly reduce immunogenicity and toxicity [17]. Amounts below 40 µg of N-CNSs did not induce toxicity in the cells tested. Because of the high capacities of loaded BiCNU by N-CNSs, we used 10 µg for subsequent experiments loaded with 10 times its weight. It is important to consider that many factors could alter the cytotoxicity induced by N-CNSs, such as size, formulation, production, and capacity to be endocytosed. The high degree of hydrophobicity of the N-CNSs allow them to be endocytosed via tip recognition through receptor binding, commonly expressed in cancer cells, including glioblastoma [18]. A possibility is that N-CNSs could interact with several biomolecules interfering with biological functions, and, in consequence, induce apoptosis [19]. The high drug-loading capacity of the N-CNSs seems to be due to the high affinity of the carbon surface to BiCNU and because the nanotubes are hollowed, allowing the retention of BiCNU over and inside the N-CNSs. Other porous carbon structures have also demonstrated high capacities to absorb hydrophobic drugs [20]; the release profile obtained using artificial CSF and DMEM support this hypothesis. The cumulative BiCNU release from N-CNSs reached 50% after the initial 5 h, and then a reduction in the rate of release was observed, which was sustained up to 72 h. This reduction in speed could be due to the liberation of BiCNU within the carbon nanotubes from the N-CNSs. The release kinetics can be diverse depending on the type of nanosponges used, and characteristics such as cavities, porous structure, size, and drug to be loaded are important features that could accelerate or prolong the release kinetics [17].

The in vitro cytotoxicity assay of N-CNSs-BiCNU was evaluated in RG2 cells at different concentrations and was compared against the cytotoxicity induced by BiCNU or N-CNSs alone at the same range of concentrations and times. Less than 10% of viability reduction was observed in the cells treated with N-CNSs without loading BiCNU, indicating a low cytotoxicity of the carrier. Both BiCNU and N-CNSs-BiCNU-treated cells showed a high toxicity toward RG2 cells with similar CC_50_ in all the times used (6.46 ± vs. 3.28 µM at 24 h; 1.6 ± vs. 1.55 µM at 48 h, and 1.36 ± vs. 2.15 ± µM respectively). These results indicate that the main cytotoxic effect was induced by the substance BiCNU, not by the carrier; thus, the carrier could deliver BiCNU effectively, without inducing a relevant cell cytotoxicity. Some studies in animal models should be performed to corroborate the induction of no relevant toxicity in the brain tissue from the N-CNSs. However, it is possible that carbon nanomaterials within the brain tissue cannot degrade and can remain at the implant site without being naturally eliminated, as we have observed in previous studies with other MWCNTs [21]. Nevertheless, N-CNSs have additional advantages with respect to the MWCNTs, as N-CNSs have presented high biocompatibilities that allow them to function as scaffolding for fibroblast growth, allowing brain plasticity [22].

## 5. Conclusions

The use of nanotechnology either as carriers for drug treatment or as a primary treatment for various diseases has been studied for several years. Our results indicate that N-CNSs may be more biocompatible than other CNTs used in similar in vitro and in vivo assays; however, more studies are necessary to elucidate whether this can be a better optimal therapeutic tool against various forms of cancer.

## Figures and Tables

**Figure 1 nanomaterials-11-01244-f001:**
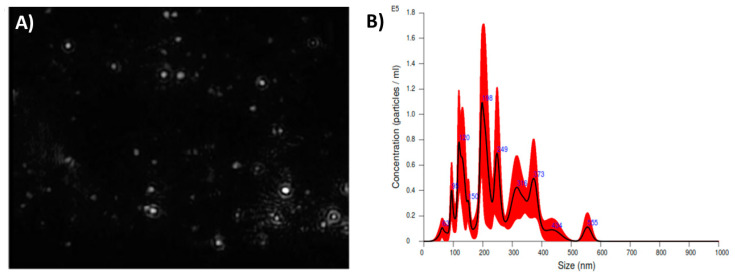
Particle size distributions (PSDs) of N-CNSs obtained by NanoSight NS300. (**A**) The NTA shows a homogeneous size distribution (nm) of N-CNSs. The experiment was run in quintuplicate. (**B**) The black line shows the mean size distribution and red line shows the error ± 1 standard error of the mean.

**Figure 2 nanomaterials-11-01244-f002:**
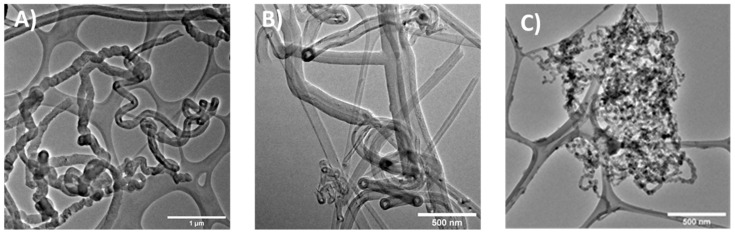
TEM images of N-CNSs: (**A**) Several types of individual CNT that constitute the N-CNSs; the structures formed from carbon nanotubes. (**B**) A T junction is clearly seen in this image. (**C**) Nanosponge built by the entanglement of CNTs that were not separated during sample preparation.

**Figure 3 nanomaterials-11-01244-f003:**
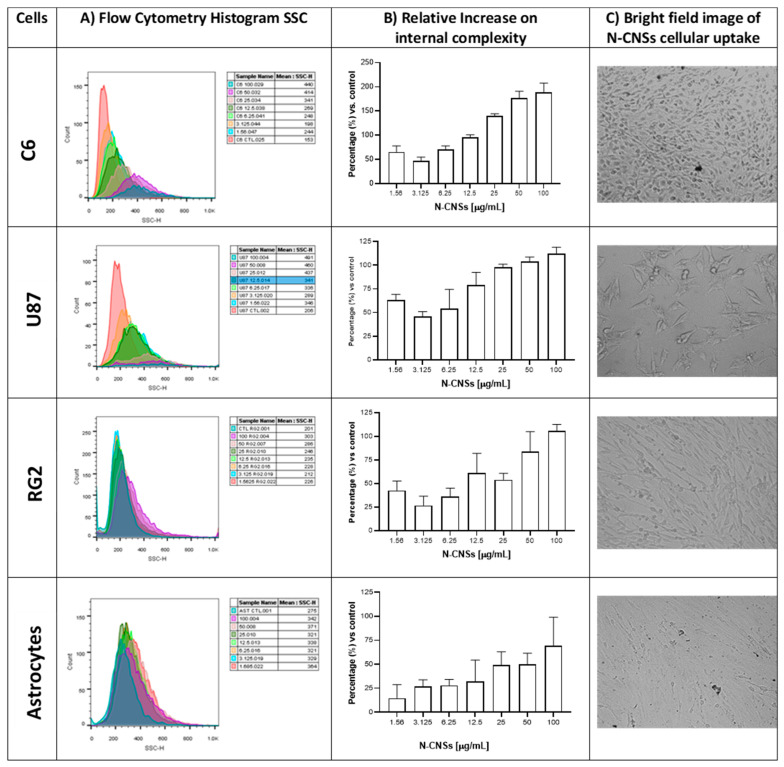
Changes in cellular granularity induced by N-CNS cell uptake. (**A**) Flow cytometry analysis of the C6, U87, RG2 cell lines, and primary astrocytes. (**B**) Relative percentage of increase in cell internal complexity in comparison to nontreated cells after incubation with N-CNSs. (**C**) Representative bright-field image of N-CNS cellular uptake after 48 h (12.5 µg/mL). Results are presented the mean ± SD of 3 independent experiments.

**Figure 4 nanomaterials-11-01244-f004:**
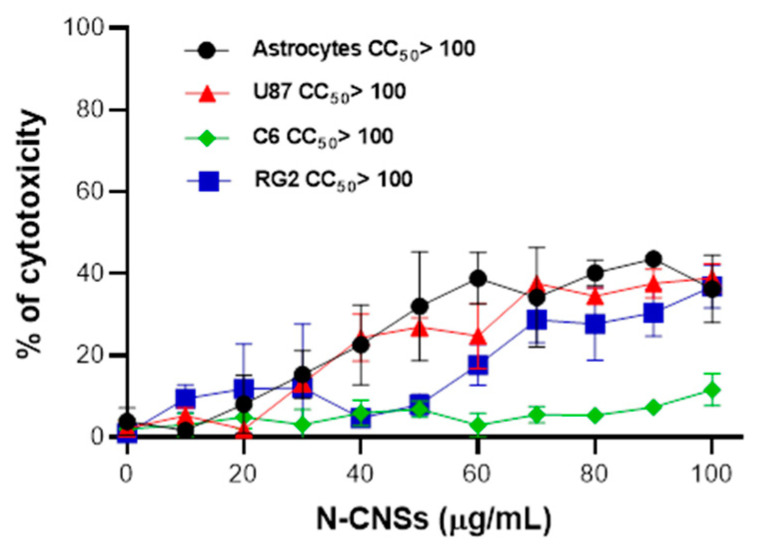
Cell viability of glioma cells treated with N-CNSs for 72 h. Astrocytes, U87, C6, and RG2 cells (1 × 10^6^) were treated with different concentrations of N-CNSs for 72 h; the percentage of dead cells with respect to the nontreated cells was determined via MTT assay. Data are shown as the mean ± SD of dead cells treated with N-CNSs. Results are presented as the mean ± SD of 3 independent experiments.

**Figure 5 nanomaterials-11-01244-f005:**
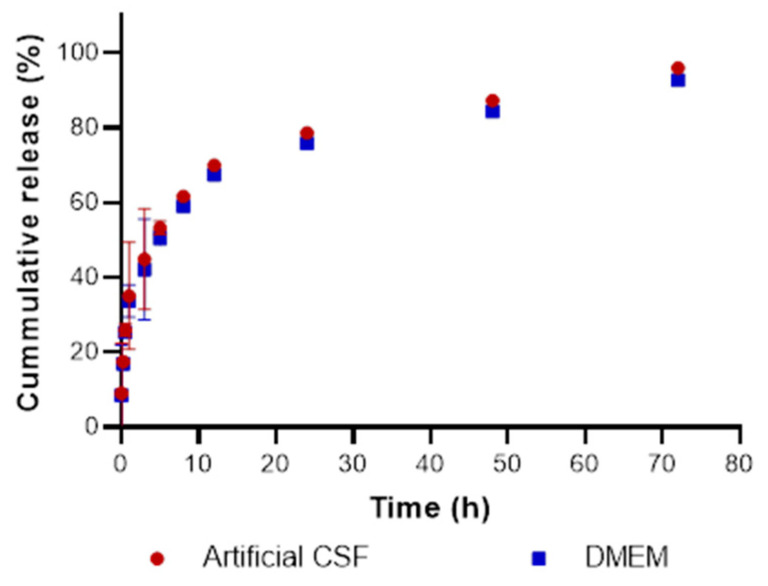
In vitro BiCNU release profiles of N-CNSs-BiCNU. Graph shows the mean ± SD of 3 different experiments by triplication of the percentage of BiCNU released after 72 h in artificial CSF (red) and DMEM (blue).

**Figure 6 nanomaterials-11-01244-f006:**
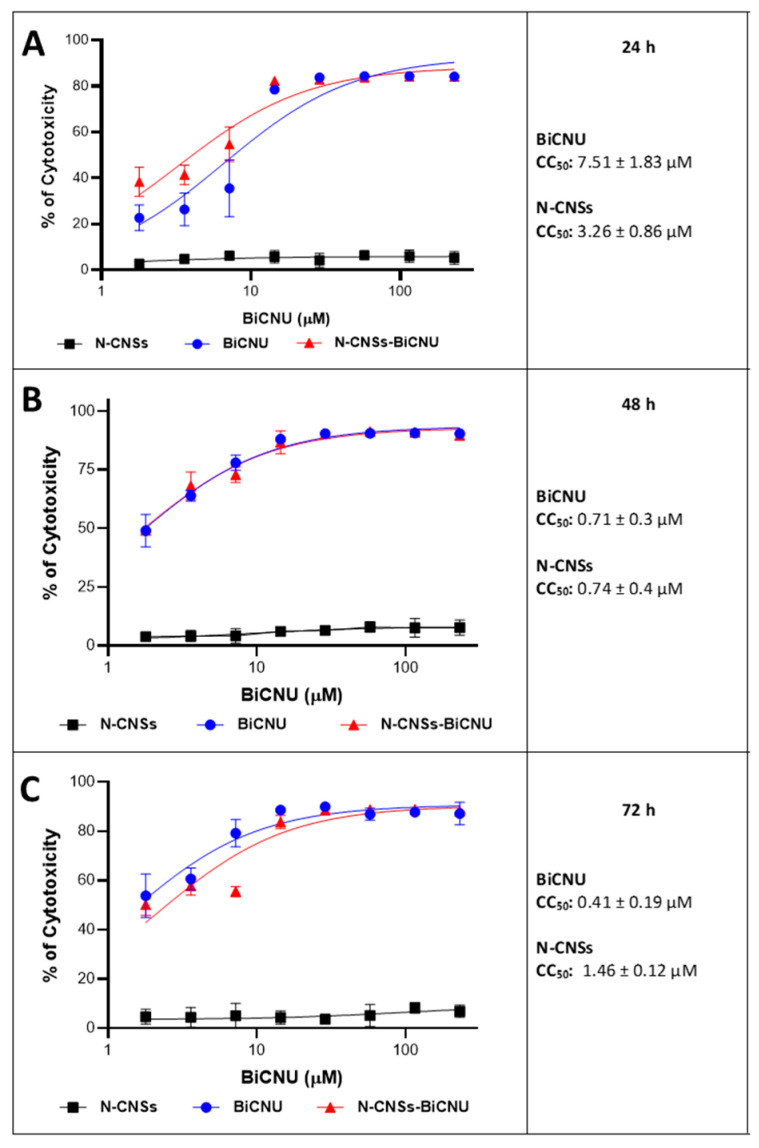
Effect of N-CNSs-BiCNU on the viability of RG2 cells at different periods. BiCNU and N-CNSs-BiCNU resulted in concentration-dependent reductions in cellular viability at 24 (**A**), 48 (**B**), and 72 h (**C**). Furthermore, the MTT assay showed that the treatment of these cells with BiCNU loaded in N-CNSs at 24 h resulted in a cytotoxic effect with a CC50 value of 3.26 ± 0.86 µM and BiCNU (7.51 ± 1.83 µM) (**A**). Similar results were observed at 48 (**B**) and 72 h (**C**) with the respective CC_50_ value. Plain N-CNSs did not induce cytotoxicity at the times tested.

## Data Availability

The data presented in this study are available on request from the corresponding author.

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
