# Peer review of "Potential Use of Nitrogen-Doped Carbon Nanotube Sponges as Payload Carriers Against Malignant Glioma"

_nanomaterials, 2021, doi:10.3390/nano11051244_

Round 1

Reviewer 1 Report

The authors have loaded their previously developed carbon nanotube sponges with a drug used in the treatment of glioblastoma. They have shown that the sponges do not cause toxicity while the drug is still able to function. This technology is an exciting concept that holds potential to impact many areas of medicine.

Please explain what you mean by ‘mass effect’. By reading the discussion section I believe that you are saying that the large size of the wafers is causing issues, but I’m not sure. Knowing this would allow the reader to understand the significance of the new carriers.

What specific type of carbon nanotubes did you use? I see that you describe how they were made, could you add a few words telling the reader what type of nanotubes they are in the methods section. The discussion section says MWCNT, but until the discussion I did not know if they were multi or single walled.

There have been reports saying that the MTT assay’s results can be altered by carbon nanotubes due to their interaction with various wavelengths of light. Is there a different way to perform your cell viability assay to back up the MTT results?

Will these nanotube sponges remain localized to the area of interest, or will they diffuse throughout the entire brain? Could you discuss your thoughts on this in the discussion?

Reviewer 2 Report

Review comments.

The paper is very important for possible cancer therapy. Some comments.

  1. Was the specific surface area (m*m/Kg) of the N-CNSs measured using the Brunauer–Emmett–Teller (BET) method or similar?
  2. In the abstract you might add the cancer drug before carmustine.
  3. Is the long term effect of the N-CNSs in the body of concern? Are the N-CNSs removed from the body in some way or could they cause additional toxicity to healthy tissue?
  4. Is the N-CNSs material acid treated or purified to remove catalyst impurities before it is used in the body? What is the amount of iron catalyst in the material? Does the iron catalyst have any toxic effects to the cells?
  5. Why are two sprayers used in the synthesis method? Can the two mixtures be injected with one sprayer?
  6. Cytotoxicity occurs due to the concentration of N-CNSs alone above 40 mg/ml. What is the mechanism of this cytotoxicity? Can the cytotoxicity be reduced by modifying the synthesis process?
  7. The diameter of the MWCNT is reported as 241.7 nm. It seems smaller diameter nanotubes would have a larger surface area than the large MWCNT used. How does the diameter of the nanotube affect the amount of drug that can be carried?
  8. It is mentioned drugs may be carried inside the CNT. Normally CNT have closed ends. Can you show drugs inside the N-CNSs?
  9. I am sorry I could not comment on the medical aspects of the cytotoxicity studies. What is the next step in testing? Mouse model studies?
  10. How are the N-CNSs delivered to the tumor? Over what period of time must the drugs be released? It would be helpful to list all the current roadblocks in treating GBM? The problems were discussed some but a list of the barriers would be helpful for non-experts to consider different solution approaches.
